# Influencing Multi-Walled Carbon Nanotubes for the Removal of Ismate Violet 2R Dye from Wastewater: Isotherm, Kinetics, and Thermodynamic Studies

**Khamael M. Abualnaja** [1]**, Ahmed E. Alprol** [2,*]**, Mohamed Ashour** [2] **and Abdallah Tageldein Mansour** [3,4,*]

1. Department of Chemistry, College of Science, Taif University, P.O. Box 11099, Taif 21944, Saudi Arabia; k.ala@tu.edu.sa
2. National Institute of Oceanography and Fisheries (NIOF), Cairo 11516, Egypt; microalgae_egypt@yahoo.com
3. Animal and Fish Production Department, College of Agricultural and Food Sciences, King Faisal University, P.O. Box 420, Al-Ahsa 31982, Saudi Arabia
4. Fish and Animal Production Department, Faculty of Agriculture (Saba Basha), Alexandria University, Alexandria 21531, Egypt
* Correspondence: ah831992@gmail.com (A.E.A.); amansour@kfu.edu.sa (A.T.M.)

**Abstract:** In this study, a multi-walled carbon nanotube (MWCNT) was synthesized and used as an adsorbent for the removal of Ismate violet 2R dye from contaminated water. The morphology and structure of the synthesized adsorbent were examined via the Brunauer–Emmett–Teller (BET) surface area, X-ray powder diffraction (XRD) analysis, infrared spectroscopy (FT-IR), scanning electron microscopy (SEM), and Raman spectroscopy. The effects of an MWCNT on the removal of IV2R were examined via a batch method using different factors such as pH, agitation time, adsorbent dosage, temperature, and initial dye concentration. The results showed that, at the acidic pH 4, 0.08 g of an MWCNT with 10 mg L$^{-1}$ at 120 min realized the favorable removal of IV2R dye using an MWCNT. Under these operation conditions, the maximum elimination efficiency for real wastewater reached 88.2%. This process benefits from the ability to remove a large amount of dye (approximately 85.9%) in as short as 10 min using 0.005 g of MWCNTs. Moreover, the investigational isotherm data were examined by different models. The equations of error functions were used in the isotherm model to show the most appropriate isotherm model. The highest adsorption capacity for the removal of the dye was 76.92 mg g$^{-1}$ for the MWCNT. Moreover, the regression data indicated that the adsorption kinetics were appropriate with a pseudo-second order and an R$^2$ of 0.999. The thermodynamic study showed that the removal of IV2R is an endothermic, spontaneous, and chemisorption process. The MWCNT compound appears to be a new, promising adsorbent in water treatment, with 91.71% regeneration after three cycles.

**Keywords:** adsorption; dye; MWCNT; kinetics; isotherm models; thermodynamic

## 1. Introduction

Aquatic pollution by industrial wastewater is a threat to human and marine lives and is a major environmental problem [1]. Industrial wastewater contains a mixture of poisonous pulp mills, surfactants, oil, organic materials, heavy metals, salts, dyestuff, and industrial discharge, as well as a massive amount of highly colored effluent, causing severe environmental concerns all over the world [2]. The dyes from textile effluents are highly resistant to pH, light, and microbial attack, which results in them remaining in the environment for a longer time [3]. Textile effluents are rich in dyes and chemical compounds, some of which are carcinogenic, non-biodegradable, toxic metals, and pose a major threat to the environment and humankind [4,5]. The majority of dyes pose a potential health hazard to all forms of life, with long-term and accidental over-exposure conditions such as eczema, skin dermatoses, and allergic responses, potentially also affecting the lungs, the liver, the immune system, the vasco-circulatory system, and the reproductive

systems of experimental animals and human systems, in addition to having a great effect on the photosynthetic activity in marine biota [6]. Therefore, various treatment methods are applied to remove color effluents and organic pollutants from wastewater, such as ion-exchange, ozonation, chemical reaction, membrane separation, filtration, adsorption, coagulation, electrodialysis, flocculation, photocatalytic degradation, reverse osmosis, and biological degradation, as well as immobilization methods [4,7–10]. These techniques have numerous disadvantages, such as chemical requirements, high energy, and low efficiency, and generally produce large quantities of sludge [11]. Although there have been improvements in several color effluent treatment technologies, the development of an effective, economic, and rapid water method at a trading level is still a defying problem. Previous efforts have concentrated on the adsorption technology for the treatment of color effluents [12,13]. This method can handle fairly high flow rates, generating high-quality wastewater that does not result in the creation of harmful materials, such as free radicals [14]. Moreover, it can minimize or eliminate various kinds of inorganic and organic contaminants, and thus has extensive applicability in the field of contamination control [15]. Furthermore, the adsorption process is successful for contaminant removal from wastewater because of its simple design, high adsorption capacity, ease of operation, insensitivity, and flexibility to toxic pollutants [16]. Over the years, several scientists and researchers all over the world have become interested in nanocompounds. The use of MWCNTs for the elimination of dye pollutant ions has recently been studied [11–15], and is comparable to additional carbon-dependent adsorbents that are utilized for commercial purposes [1].

Comparatively, MWCNTs, as newly developed carbonaceous substances, have received significant interest for application as potential adsorbents in wastewater treatment because of their distinct structure, shorter equilibrium time, great specific surface area, higher sorption capacity, flexible surface chemistry, and lower mass loss in reactivation [17,18]. MWCNTs have been confirmed to be more efficient for environmental contamination control in terms of the elimination of organic materials such as dyes from aquatic effluents, as reported by several researchers such as Machado et al. [19], Mashkoor et al. [20], and Yao et al. [21]. In addition, Shirmardi et al. [22] studied the elimination of malachite dye from aquatic solutions using MWCNTs. The results were found to be $q_{max}$, 142.82 mg g$^{-1}$ of dye at 80 min, and pH 7. Recently, Alkaim et al. [23] synthesized carbon nanocompounds for the removal of malachite dyes in aqueous solutions. The maximum capacity of the adsorption process was observed to be 187.69 mg g$^{-1}$ at 120 min and pH 10. Machado et al. [24] studied the adsorption of a reactive blue 4 dye from an aqueous mixture by carbon nanotubes. The maximum sorption capacity was observed to be 502.5 mg/g. Moreover, Saber-Samandari et al. [18] studied the adsorption of anionic and cationic dyes from a liquid mixture using gelatin-based magnetic nanocomposite beads, including carboxylic acid functionalized carbon nanotubes. The results showed the percentage removal to be 76.3% of methylene blue. Gang Yu et al. [25] reported that the treatment of organic chemicals compounds by MWCNTs may include more mechanisms, such as hydrophobic interactions, π–π interactions, electrostatic interactions, Lewis acid–base interactions, and hydrogen bonding, which reveals that this requires essential additional investigation.

Ismate violet 2R was selected as a model compound in this research due to its wide application range, which includes coloring paper, and dyeing cottons, wools, and coating for paper stock and medical purposes, in addition to its potential dangerous effects.

Accordingly, the aims of this work were the preparation, functionalization, and purification of multi-walled carbon nanotubes (MWCNTs), as a novel approach, and the application of MWCNTs as an adsorption substance to remove Ismate violet 2R (IV2R) dye from aquatic effluents under various conditions using the batch adsorption method. The main parameters evaluated were the initial dye concentration, pH, contact time, temperature, and adsorbent dose. Moreover, the changes in the physical, chemical, and morphological structure owing to imitation reactions were studied through Raman, XRD,

BET surface, FTIR, and SEM characterizations. Additionally, thermodynamic and kinetics studies were conducted, and the experimental equilibrium was investigated using various sorption isotherm models to assess the adsorption mechanism. In addition, to estimate the regenerative capacity of MWCNTs, the interference of wastewater on the adsorption of IV2R dye was also examined. The novelty of this study can be summarized as the first case, in the oxidized MWCNT literature, of MWCNTs modified with such functionalities and applied to the elimination of IV2R dye. Furthermore, previous studies on dye removal by a single functional group indicated lower elimination than that presented in this work. Moreover, the highest suitable model was achieved by using error functions.

## 2. Materials and Methods

### 2.1. Materials

MWCNTs were created using the chemical vapor deposition technique, in which acetylene with cobalt and an iron solution was placed in an inert gas atmosphere connected to a reaction chamber. In this process, nanotubes were made on the substrate through the decay of hydrocarbon at an atmospheric pressure of 600–900 °C. This method can be scaled up to prepare, purify, and functionalize MWCNTs according to Mohammed et al. [26] and Bahgat et al. [27], who reported the synthesis of MWCNTs via the following steps: 2 g of the supporting catalyst was injected into an alumina boat and displayed on a cylindrical quartz tube fitted inside a furnace at 600 °C, and the catalyst was heated in the presence of $N_2$ gas for 10 min with a rate of 90 mL min$^{-1}$. The movement of acetylene gas was tolerated by the quartz tube undergoing catalyzation at 90 mL min$^{-1}$ with a flow rate of 40 min after the catalyst was heated. The flow of acetylene ceased after the required period, and the new product was cooled at room temperature.

### 2.2. Purification and Functionalization of MWCNTs

An enormous surface area prompts a strong tendency to shape agglomerates. Surface functionalization helps in steadying the scattering, since it can repress the reaggregation of nanotubes and can prompt the coupling of polymeric grids with MWCNTs. Covalent functionalization of MWCNTs can be determined by identifying some functional groups on sites of MWCNTs through oxidizing agents, such as strong acids, which causes the formation of hydroxyl or carboxylic groups (–OH, –COOH) on the outside of the nanotubes. Such groups are known as functionalization-type defect groups. The functionalization process was performed using 10 mL of concentrated sulfuric acid and 30 mL of nitric acid in a 250 mL flask loaded with 10 g of the produced MWCNTs and 5 g of phosphorous pentaoxide. The mixture solution was refluxed for 120 min at 350 °C to obtain an MWCNT suspension mixture. Then, the mixture solution was washed with deionized water, followed by drying for 24 h at 50 °C to obtain carboxylate MWCNTs [27].

### 2.3. Dye Solution Preparation

Stock solutions of IV2R were set by dissolving accurately weighed dye in distilled water at a concentration of 1 L, without additional purification. The characteristics of the dye used in this study are presented in Table 1. The experimental mixtures were achieved by diluting the dye stock mixture in accurate proportions to obtain various initial concentrations.

### 2.4. Adsorption Experiments

The effects of pH (2–10), contact time (10–180 min), initial dye concentration (10–80 mg L$^{-1}$), adsorbent dose (0.005–0.08 g), and temperature (25–55 °C) for IV2R removal were examined. The particular conditions are cited in the related plots. The tests were conducted in triplicate by the batch technique to gather equilibrium data. The reaction mixture containing 50 mL of the dye solution and the solution were shaken at room temperature (25 ± 2 °C) at a speed of 110 rpm under fixed conditions in an incubator shaker. The pH value was adjusted by HCl or NaOH before adding the adsorbent. After shaking, the concentration of

IV2R dye was determined in a clear supernatant at any time at 550 nm using a UV–visible spectrophotometer. Concentrations of dye in a solution can be assessed quantitatively, as stated by the Lambert–Beer law, by linear regression equations, achieved through plotting a calibration curve for the concentration ranges of dye.

**Table 1.** Chemical and physical characteristics of the ISMATE violate 2R dye [28].

| Characteristics | Value |
|---|---|
| Dye name | ISMATE violate 2R |
| Wavelength ($\lambda$ max) | 550 nm |
| Mol. wt. | 700 |
| Molecular formula | $C_{22}H_{14}N_4O_{11}S_3CuCl$ |
| C.I. name | IV2R |
| Molecular structure |  |

The adsorption capacity $(q_e)$ and dye elimination percentage were calculated using Equations (1) and (2), as follows [29]:

$$q_e = \frac{\left(C_i - C_f\right) \times V}{W} \tag{1}$$

$$\text{Percentage removal } (\%) = \frac{\left(C_i - C_f\right)}{C_i} \times 100 \tag{2}$$

where $C_i$ and $C_f$ (mg L$^{-1}$) are the primary concentration at the initial time and the final concentration of IV2R at a certain period, respectively, while $V$ represents the volume of the dye mixture (L) and W relates to the weight of the dry adsorbent (g).

*2.5. Characterization of MWCNTs*

The morphologies of the adsorbents were characterized using a scanning electron microscope (JEOL JSM 6360 LA). Moreover, Brunauer–Emmett–Teller (BET) desorption–adsorption tests were conducted on a BELSORP mini-II, BEL Japan, Inc. The specific external area, mean pore diameter, and pore volume of the MWCNTs were calculated by means of $N_2$ desorption–adsorption measurements at an $N_2$ solution with an 89.62 kPa saturated vapor pressure and an adsorption temperature of 77 K.

Furthermore, FTIR and Raman analysis spectrophotometry were used to measure the influence of the prepared MWCNTs materials—a Shimadzu FTIR-8400 S (Japan) and a Bruker Senterra Raman spectrometer were utilized, respectively. Additionally, the samples were assessed using a particle size analyzer (Beckman Coulter; Miami, FL, USA) at angles of 11.1° and 90° to determine the distribution of particle sizes. Additionally, an X-ray diffraction apparatus (D8 Advance X-Ray Di-ractometer; Bruker AXS, Wisconsin, USA) was used to investigate the crystal structure of the MCNTs with Panalytical in a scope from 10° to 90° with K$\alpha$ Cu radiation.

*2.6. Adsorption Isotherm Study*

2.6.1. Isotherm Experiment

The extent of removal of IV2R from aqueous solutions depends heavily on the initial dye concentration. In order to assess isotherm study, different IV2R concentrations varying from 10 to 100 mg L$^{-1}$ were examined at constant parameters and pH 6 by 0.01 g of MWCNTs adsorbents, at 30 °C, for 3 h, at 150 rpm, and were mixed with 50 mL of dyes solution [30]. The data were fitted and calculated in terms of the following isotherms:

2.6.2. Theoretical Background of Isotherm Models

The Freundlich Model

The ability of the Freundlich model to fit the experimental data, via plotting a curve of log $q_e$ with respect to log $C_e$, was employed to generate a slope of n and the intercept value of $K_f$. The Freundlich model can be easily linearized by plotting it in a logarithmic form [31]:

$$\log q_e = \log K_f + 1/n \log C_e \tag{3}$$

Freundlich constants $K_f$ and *n* were determined from the isotherm equation according to Equation (3).

The Langmuir Model

The Langmuir model is presented by mathematical expression [32] as the following equation:

$$q_e = q_{\max} bC_e/(1 + bC_e) \tag{4}$$

where $q_{\max}$ (mg g$^{-1}$) is the maximum sorption capacity corresponding to the saturation capacity (representing the total binding sites of a fiber) and *b* (L mg$^{-1}$) is a coefficient related to the affinity among the MCNTs and dye ions.

The linear relationship can be achieved from plotting curve $(1/q_e)$ vs. $(1/C_e)$:

$$1/q_e = 1/(bq_{max} C_e) + 1/q_{max} \tag{5}$$

in which *b* and $q_{\max}$ are determined from the slope and intercept, respectively.

The Henderson and Halsey Isotherm Models

These models are suitable for heteroporous solids and the multilayer sorption technique [33,34]. The Halsey model was calculated using the following equation:

$$\ln q_e = \frac{1}{n}\ln K + \frac{1}{n} \ln C_e \tag{6}$$

where *n* and *K* are Halsey constants. Meanwhile, the Henderson model was obtained from the following equation:

$$\ln[-\ln(1 - C_e)] = \ln K + \left(\frac{1}{n}\right) \ln q_e \tag{7}$$

while the Henderson constants are *nh* and *Kh*.

The Harkins–Jura Model

This model explains multilayer sorption, in addition to the presence of heterogeneous pore scattering in a sorbent [34]. This model was obtained from the following equation:

$$\ln q_e = \frac{1}{n}\ln K + \frac{1}{n} \ln C_e \tag{8}$$

where the isotherm constants are *AHJ* and *BHJ*.

The Smith Model

The Smith model is appropriate for heteroporous solids and multilayer adsorption. This model is usually obtained from the following equation:

$$q_e = W_{bS} - W_S \ln(1 - C_e) \tag{9}$$

where the Smith model parameters are $W_{bS}$ and $W_s$.

The Tempkin Model

The Tempkin isotherm model assumes that the heat of the sorption of each particle in the layer declines with coverage because of adsorbate–adsorbent interactions; moreover, the sorption process is described through a uniform distribution of the binding energies, up to some higher binding energy [35]. The Tempkin model was calculated from the following equation:

$$q_e = B \ln A + B \ln C_e \tag{10}$$

where $A$ is the constant of equilibrium binding (L g$^{-1}$) associated with a higher binding energy, $b$ (J mol$^{-1}$) is a constant corresponding to the heat of sorption, and $B = (RT b^{-1})$ (J mol$^{-1}$) is the Tempkin constant and is related to the heat of adsorption and the gas constant (R = 8.314 J mol$^{-1}$ K$^{-1}$).

2.6.3. Error Functions Test

To determine the greatest and most appropriate model to investigate the equilibrium data, various error functions were studied. The following models were used for the error functions test [36].

Average Percentage Error (ABE)

The ABE model shows a tendency or appropriateness among the predicted and experimental values of the sorption capacity used for plotting model curves (ABE) and can be designed consistent with the following equation:

$$\text{APE}(\%) = \frac{100}{N} \sum_{i=1}^{N} \left| \frac{q_{e,isotherm} - q_{e,calc}}{q_{e,isotherm}} \right|_i \tag{11}$$

where $n$ shows the number of investigational data points.

Nonlinear Chi-Square Test ($\chi^2$)

The nonlinear chi-square test is a statistical approach for the best appropriate treatment system. The approach of the chi-square error, $\chi^2$, model is assumed as the following equation:

$$X^2 = \frac{(q_{e,isotherm} - q_{e,calc})^2}{q_{e,isotherm}} \tag{12}$$

Sum of Absolute Errors (EABS)

An augmentation in errors provides a better fit, leading to a bias toward great concentration data. EABS tests can be assessed using the following equation:

$$\text{EABS} = \sum_{i=1}^{N} \left| q_{e,calc} - q_{e,isotherm} \right|_i \tag{13}$$

*2.7. Adsorption Kinetics*

Adsorption kinetics studies were conducted in 50 mL conical flasks. A solution of pH 2 with 0.01 g of adsorbent was mixed separately with 50 mL of the IV2R solution of 10 mg L$^{-1}$ concentrations, and the solution was agitated at room temperature under

requisite time intervals viz. 10, 20, 30, 60, 120, and 180 min [37]. The clear solutions were analyzed for residual IV2R concentrations in the solution.

### 2.7.1. Pseudo-First-Order Kinetics Model

The linear form of the generalized pseudo-first-order equation [38] is given by the following equation:

$$d_q/d_t = K_1 (q_e - q_t) \tag{14}$$

where $q_e$ is the amount of dye adsorbed at equilibrium (mg g$^{-1}$), $q_t$ is the quantity of dyes adsorbed at time $t$ (mg g$^{-1}$), and $K_1$ is expressed as a pseudo-first-order rate constant (min$^{-1}$). The integrating equation was assessed as follows:

$$\text{Log } (q_e/q_e - q_t) = k_1 t/2.303 \tag{15}$$

The pseudo-first-order equation is given by the following formula in a linear equation:

$$\text{Log } (q_e - q_t) = \log q_e - k_1 t/2.303 \tag{16}$$

Plots of $\log(q_e - q_t)$ against ($t$) should provide a linear relationship from $k_1$ and $q_e$, which can be assessed via the slope and intercept, respectively [39].

### 2.7.2. Pseudo-Second-Order Kinetics Model

The pseudo-second-order equation is expressed as follows [40]:

$$dq_t/d_t = K_2(q_e - q_t)^2 \tag{17}$$

where $K_2$ indicates the constant of the second-order rate (g mg$^{-1}$ min). The integrated equation is presented as follows:

$$1/(q_e - q_t) = 1/q_e + K_2 \tag{18}$$

Ho et al. [11] obtained a linear form of the pseudo-second-order equation as follows:

$$t/q_t = 1/K_2 q_e{}^2 + t/q_e \tag{19}$$

Plots of ($t/q_t$) against ($t$) would provide a linear relationship, and the values of the $q_e$ and $K_2$ parameters can be calculated from the slope and intercept, respectively.

### 2.7.3. The Intraparticle Diffusion Model

The intraparticle diffusion equation [14] is as follows:

$$q_t = K_{dif} t^{1/2} + C \tag{20}$$

where $C$ is the intercept and $K_{dif}$ (mg g$^{-1}$ min$^{-0.5}$) reflects the intraparticle diffusion rate constant, which is estimated through the slope of the regression line.

### 2.8. Adsorption Thermodynamics

A thermodynamic study is necessary for taking information about whether the nature of sorption techniques is spontaneous or not, and the Gibbs and the value of free energy change, $\Delta G^\circ$, are important principles of non-spontaneity. The changes in the enthalpy $\Delta H^\circ$ and entropy $\Delta S^\circ$ parameters should be considered to calculate the Gibbs free energy for sorption techniques at various temperatures (e.g., 25, 35, 45, and 55 °C) using the following nonlinear forms [41]:

$$K_d = q_e/C_e \tag{21}$$

$$\Delta G^\circ = -RT \ln K_d \tag{22}$$

$$\Delta G^\circ = \Delta H^\circ - T \Delta S^\circ, \text{ or} \tag{23}$$

$$\Delta G° = T (\Delta S°) + \Delta H° \tag{24}$$

The values of $\Delta H°$ and $\Delta S°$ were calculated from the intercept and slope of the plotted curve of T vs. $\Delta G°$ of Equation (23) or (24), in addition to $\Delta G°$, which can be obtained from Equation (22).

### 2.9. Application on Real Wastewater

A wastewater sample was collected from the El-Emoum drain, which neighbors Lake Maruit, Alexandria, Egypt, and which comprises numerous industrial sewage and agricultural wastes. The wastewater had a pH of 9.5, a TSS of 250 mg L$^{-1}$, and a TDS of 3240 mg L$^{-1}$. Owing to the low concentrations of dyes, an amount of IV2R dye was added to obtain 10 mg L$^{-1}$ of this dye. The dye solution was filtered to remove precipitates and suspended matters. The application of MWCNTs to remove the IV2R dye from the wastewater was carried out under optimized conditions (pH 2 for 120 min at 30 °C and 0.08 g of adsorbents in the solution). Deionized water comprising a similar concentration of dyes was prepared to use as a control to estimate the effect of adsorbent on IV2R dye removal. The absorbance was determined using a Schimadzu UV 2100 spectrophotometer. Furthermore, the pH was determined by a digital electrode pH meter, while the total suspended solids (TSS) and total dissolved solids (TDS) were measured by standard methods [42].

## 3. Results and Discussion

### 3.1. Characterization of the MWCNTs

#### 3.1.1. FT-IR Analysis

Infrared spectroscopy is a useful qualitative tool for determining the composition of compounds. Figure 1 shows the typical FTIR spectra in the range of 500–4000 cm$^{-1}$ for MWCNTs. In the spectra of the functionalized MWCNT material, the broad peaks at 3510–3780 cm$^{-1}$ are assigned to N-H and O-H stretching of the carboxylic group, respectively. The increased intensity of the OH stretching broadband at ~3510 cm$^{-1}$ suggests a greater proportion of defective CNTs in the sample, which can be easily oxidized in the air [43]. However, the band at 2358 cm$^{-1}$ is recognized as thiol S-H stretching. The sharp band at 1730 cm$^{-1}$ corresponds to C-O stretching vibration of COOH. The FTIR of amine-functionalized CNT shows new bands at 1654 cm$^{-1}$ and corresponds to C=C stretching, C=N due to the reaction with hydroxyl amine, and N-H deformation [44]. The difference in the FTIR spectra of the MWCNTs in the 1000–1270 cm$^{-1}$ region can be attributed to C-N stretching due to carbon atom and terminal amino group bonding, and confirms the presence of amine groups on the MWCNTs [44].

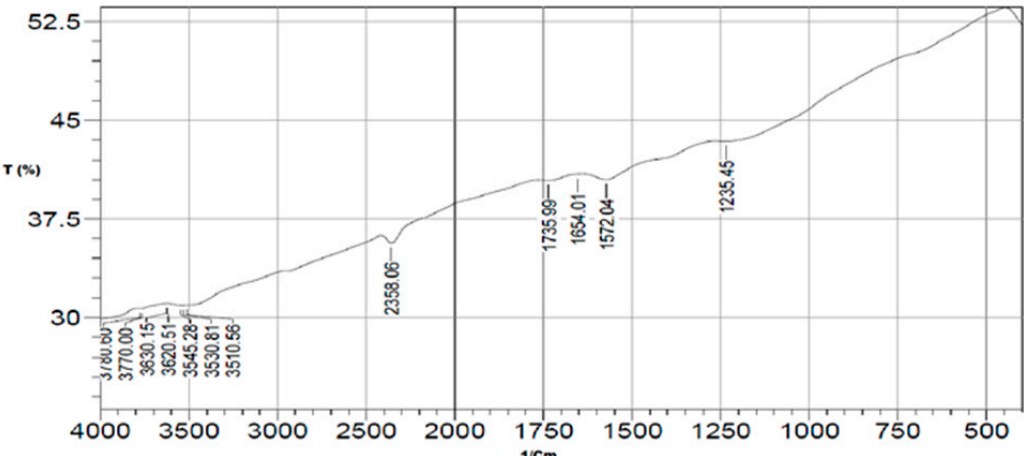

**Figure 1.** FTIR pattern of MWCNTs.

### 3.1.2. BET Surface Analysis

The BET and nitrogen adsorption–desorption study investigation method was utilized to quantify the pore diameter and specific surface space of the MWCNTs (Figure 2). A sample of the MWCNTs was optimized at a deposition temperature of 750 °C. The results of the BET surface area examination of the MWCNTs show that the pore diameter was 6.67 nm, and the whole surface area was 181.99 $m^2 \, g^{-1}$. The surface area of CNTs is affected by their diameter, agglomeration, surface functionalization, presence of impurities, and so on [45].

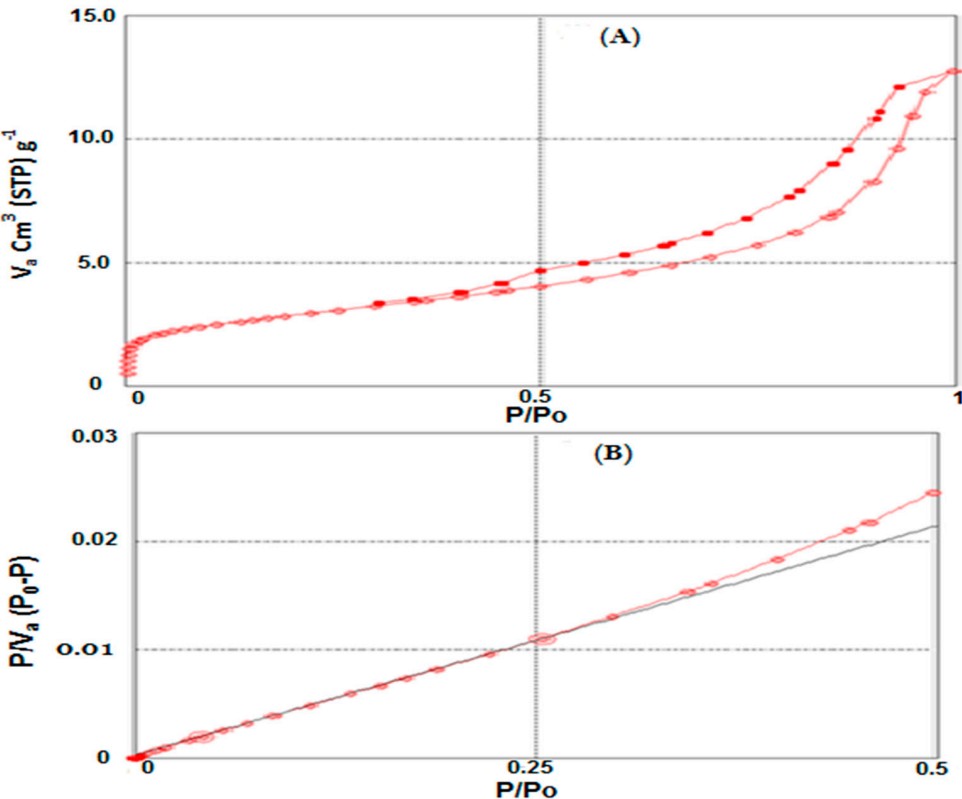

**Figure 2.** BET specific surface area (**A**) and adsorption–desorption isotherm examination (**B**) of the MCNTs.

### 3.1.3. SEM Examination

The external morphology of the MWCNTs was examined by scanning electron microscopy. Figure 3A,B displays the SEM images as prepared and purified MWCNTs, respectively. These figures show greatly tangled tubes and the existence of amorphous carbon in the unpurified sample. This observation, after acid treatment, confirms the disappearance of the amorphous texture, and there is no evidence of damage to the tubular structure of the MWCNTs, nor did the multi-walled arrangement vary. The tubular diameter of the MWCNTs ranged between 75 and 100 nm.

### 3.1.4. XRD Measurement Analysis

X-ray diffraction was applied in order to investigate the changes in the crystalline structure of the MWCNTs. Figure 4 shows the XRD patterns of the MWCNTs. It is clear from the figure that there are strong bands at 20 at 28.08°, 31.4°, 34°, and 44.66° for the oxidized MWCNTs, corresponding to an interplanar space of 3.41 Å and 2.05 Å, respectively. This could be due to the (002) and (100)/(101) planes of the MWCNTs (graphite structure) providing improved crystallinity and a decline in the amorphous structure due to acid treatment. Similar observations have been described for MCNTs prepared via the CVD method [46,47].

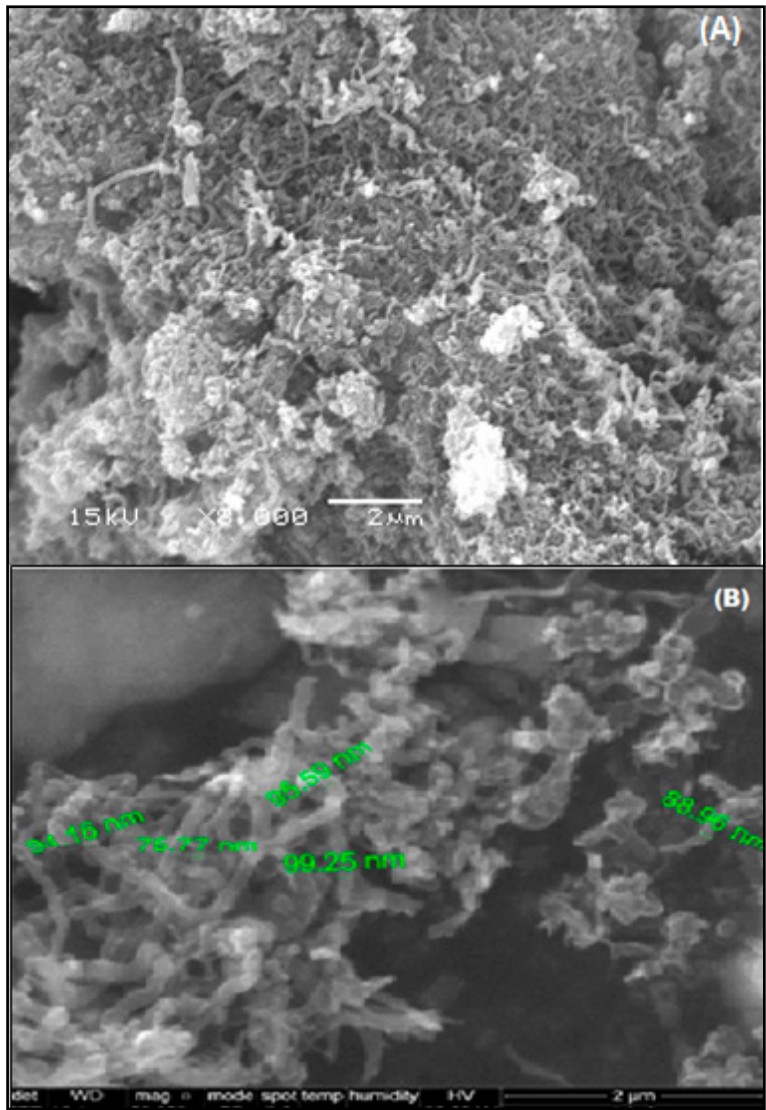

**Figure 3.** SEM examination at 2 μm and magnification (8000×) for the prepared (**A**) and purified (**B**) MWCNTs.

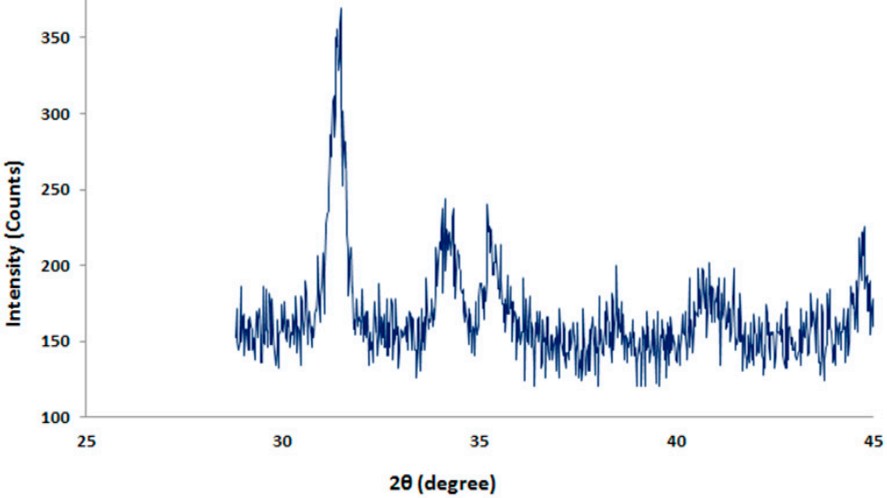

**Figure 4.** XRD diffraction of the MWCNTs.

### 3.1.5. Raman Characterization

A Raman spectrum displays different peaks that undergo alterations with changes in the characteristics of a sample to identify the chemical composition or microstructure constituents of the sample. A Raman spectrum of the MWCNTs is represented in Figure 5. The peak intensity ranging from ~1300 to 1350 cm$^{-1}$ represents disordered carbon (D band), while peaks ranging from ~1580 to 1600 cm$^{-1}$ represent graphitic carbon (G band). The D peak for the MCNTs occurred at ~1325 cm$^{-1}$, while G peaks were observed at ~1580 cm$^{-1}$. The ratio of ID/IG was calculated to estimate the variation of CNTs' quality and quantity of different defects. The ratio for the CNTs was 0.83; a smaller ID/IG ratio (<1.0) indicates a higher quality of CNTs [48]. A sample has quite a high number of defects if the intensity of these peaks is proportional [49]. Additionally, the peaks at 10–200 cm$^{-1}$ are assigned to the lattice vibrations in crystals. The absorption peaks at 640, 830, 1080, and 1450 cm$^{-1}$ are attributed to the $\upsilon$(C-S), $\upsilon$(C-O-C), $\upsilon$(C=S), $\delta$(CH$_2$), and $\delta$(CH$_3$) asymmetric, respectively. Furthermore, the absorption peak at 1630 cm$^{-1}$ was caused by –C=N group stretching. The absorption band at 1806 cm$^{-1}$ is assigned to the C=C group. The carbonyl group was found in 2000 cm$^{-1}$. However, the peaks at 3150 and 2570 cm$^{-1}$ are assigned to $\upsilon$ (O-H) and $\upsilon$ (-S-H), respectively.

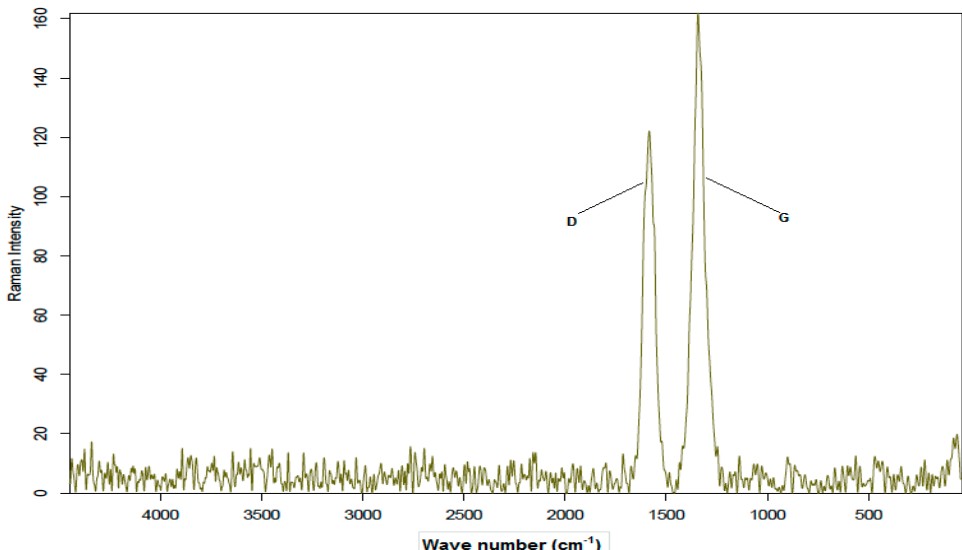

**Figure 5.** Raman spectrum for the MWCNTs.

### 3.2. Adsorption Experiments

### 3.2.1. Influence of pH

The pH of a solution may change the form and chemistry of the target dye ions and the binding sites on the adsorbent. The effect of pH on the removal of IV2R ions by MCNTs was performed in a pH range of 2–10, the results of which are shown in Figure 6. These results show that when the pH of the solution was augmented from 2 to 4, the adsorption of dye increased to reach its maximum for the MCNTs of 92.98% at pH 4; then, it slightly decreased. Therefore, pH 4 was selected as a compromise value. This can be explained by the adsorbent containing several functional groups that are affected by the pH of the solution. Moreover, an acidic solution increases in the percentage of anionic dye removal because of the electrostatic attraction between anionic dye and the positive charge of the surface of the adsorbent [50]. At a higher pH, electrostatic repulsion occurs among dye molecules and the negatively charged surface, therefore reducing the percentage elimination of anionic dyes and the sorption capacity [11]. Additionally, Ghoneim et al. [29] concluded that, at a higher pH, elimination is low compared to the maximum condition. This can be clarified as the binding site not being able to activate under basic conditions.



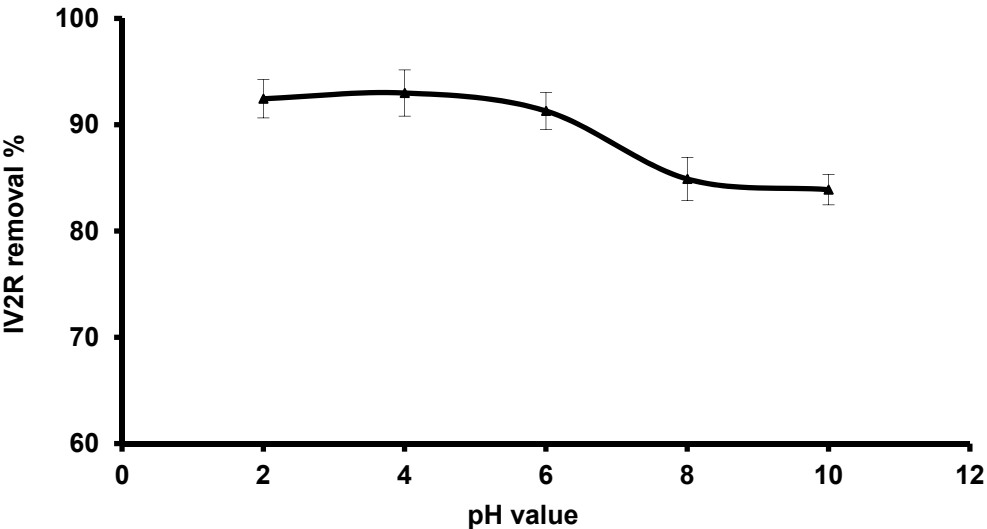

**Figure 6.** Influence of the pH value on the sorption of IV2R dye.

### 3.2.2. Influence of Adsorbent Dose

The mass of an adsorbent shows its capacity for a given concentration of dye. The influence of the adsorbent dose was studied at 25 °C by varying the sorbent dose to 0.005, 0.01, 0.2, 0.04, and 0.08 g. For all of these runs, the concentration of the IV2R dye was fixed at 10 mg L$^{-1}$, and the results are illustrated in Figure 7. An increase in the percentage of removal was observed when the dose was increased from 0.005 to 0.08 g, and the maximum removal was obtained at the adsorbent dose of 0.08 g with a percentage of 98.32% for the IV2R dye in the MWCNTs. The obtained data show that the adsorption of dye increased gradually with an augmentation in the amount of MWCNTs due to the greater availability of dye-binding sites of the surface area at higher concentrations of the adsorbent [51]. Zhao et al. [17] studied the treatment of methyl orange dye by MWCNTs, and it was found that, as the mass of MWCNTs increases, the whole adsorptive capacity of MO increases because of the increase in the number of reaction sites available and surface area to MO.

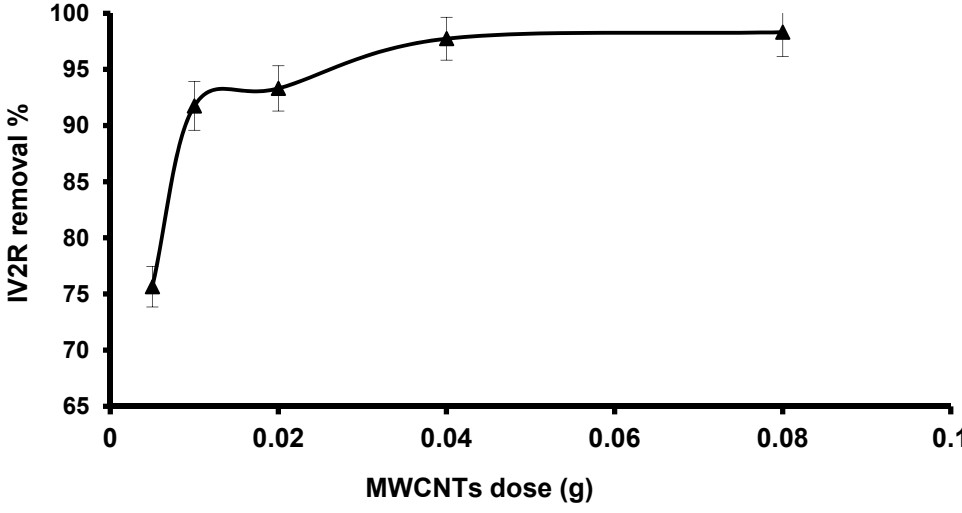

**Figure 7.** Influence of the MWCNT dose on the removal method.

### 3.2.3. Influence of Contact Time

Contact time is an essential factor in all transfer phenomena for the adsorption process, and the equilibrium time is important when considering economical water in addition to wastewater applications. The results are presented in Figure 8, indicating that the adsorption of IV2R dye ion removal increased to reach equilibrium, with 91.67% at 120 min.

The adsorption was rapid at the first time of contact, which occurred in the early stage of adsorption after a few minutes; this was due to the fact that most of the binding sites were free and the adsorbent sites were empty [52], which allowed the quick binding of dyes ions on the adsorbent. Adsorption happens rapidly and is usually controlled via the diffusion method from the bulk of the solution on the surface [53]. The increasing contact time increases the adsorption, which is possible because of a greater surface area of the MWCNTs being available at the beginning and exhaustion of the conversion of external adsorption sites, at which the adsorbate (dye particles) is transported from the external to the internal sites of the MWCNTs adsorbent molecules [54]. Hashem [55] recorded that adsorption occurs in the first 40 min of the sorption method, which shows the high diffusion of the dye particles into the outer surface of the adsorbent. In later stages, fewer vacant adsorption sites are converted, which reduces the rate of adsorption, in addition to reaching an equilibrium state.

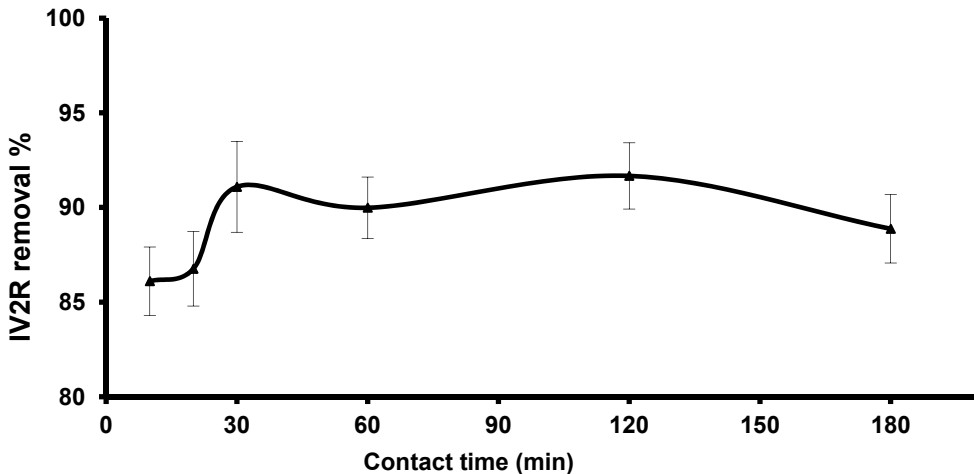

**Figure 8.** Influence of agitation time on the removal of IV2R.

### 3.2.4. Influence of the Initial Dye Concentration

The extent of removal of dyes from an aqueous mixture depends heavily on the dye concentration. The influence of the initial concentration of dye ions in a series of 10–70 mg L$^{-1}$ on adsorption was examined, and the results are shown in Figure 9. By using MWCNTs, the maximum uptake was found at the concentration of 10 mg L$^{-1}$ (91%) for IV2R dye. After this, the dye removal rate decreased with the rising concentration of IV2R dye until the end of the experiments, at the same contact time and adsorption temperature. However, the results showed that the amount of dye adsorbed per unit mass of adsorbent increased from 1.2 to 5.5 mg g$^{-1}$ when the concentration of dye increased from 10 to 70 mg L$^{-1}$. The increase in the adsorption capacity is possibly due to greater interaction among the dye and adsorbent, in addition to an increase in the driving force of the concentration gradient with the increase in the initial concentration [17]. Moreover, the higher quantity of dye removal at higher concentrations is probably because of increased diffusion and decreased resistance to dye uptake [56]. Furthermore, the rapid initial stage of dye removal with this adsorbent is ascribed to the availability of active sites on the adsorbent and to the gradual occupancy of these sites; the sorption converts to be less efficient. Generally, increasing the dye concentration leads to a decrease in the percentage of removal of dye, which may be because of the saturation of sorption sites on the surface of the adsorbent [1,39]. Moreover, the major quantity of dye adsorbed at a high dye concentration could be related to two main effects, namely, the high probability of collision between dye ions with the adsorbent's surface and the high rate of dye ion diffusion onto the adsorbent's surface [57].

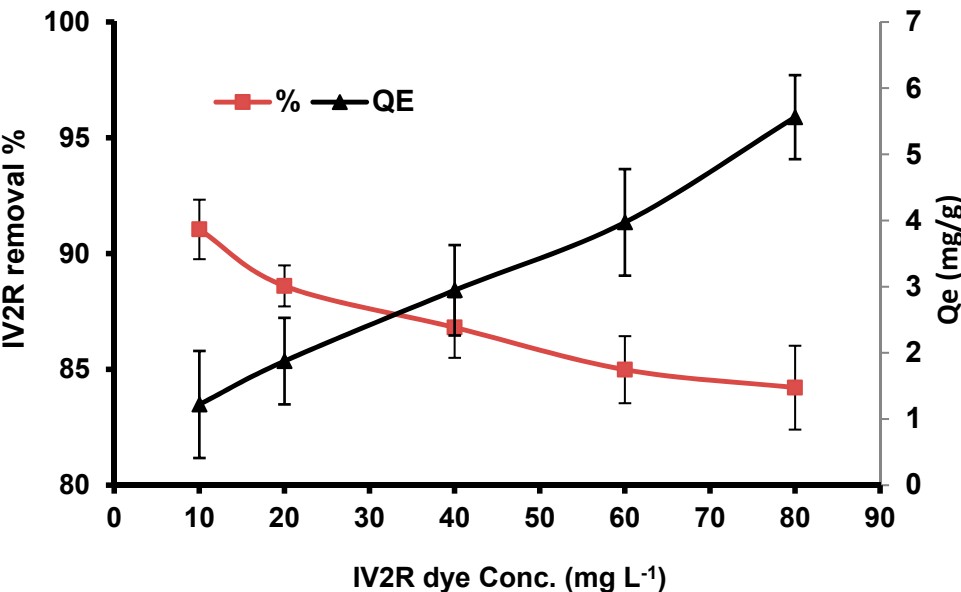

**Figure 9.** Influence of the initial IV2R dye concentration.

### 3.2.5. Influence of Temperature

Figure 10 shows the variation of the percentage of removal of IV2R dye ions in the MWCNTs. It can be concluded that the maximum percentage of removal of dye was obtained at 30 °C with 91.15% and reduced with the rise in temperature from 30 to 55 °C for the IV2R dye in the MWCNTs. With a rise in temperature from 25 to 30 °C, the diffusion rate of the adsorbate molecules within the pores, because of the decreasing solution viscosity, also modified the equilibrium capacity of the MWCNTs for a particular adsorbate. A further increase in temperature (above 30 or 35 °C) led to a decrease in the percentage removal. This is mainly due to the decreased surface activity [58,59]. This decrease in the efficiency of the adsorption may be ascribed to several causes, such as deactivation of the adsorbent's surface, a growing tendency for the dyes to escape from the solid stage to the bulk stage, or the destruction of specific active sites on the surface of the adsorbent owing to bond ruptures [16].

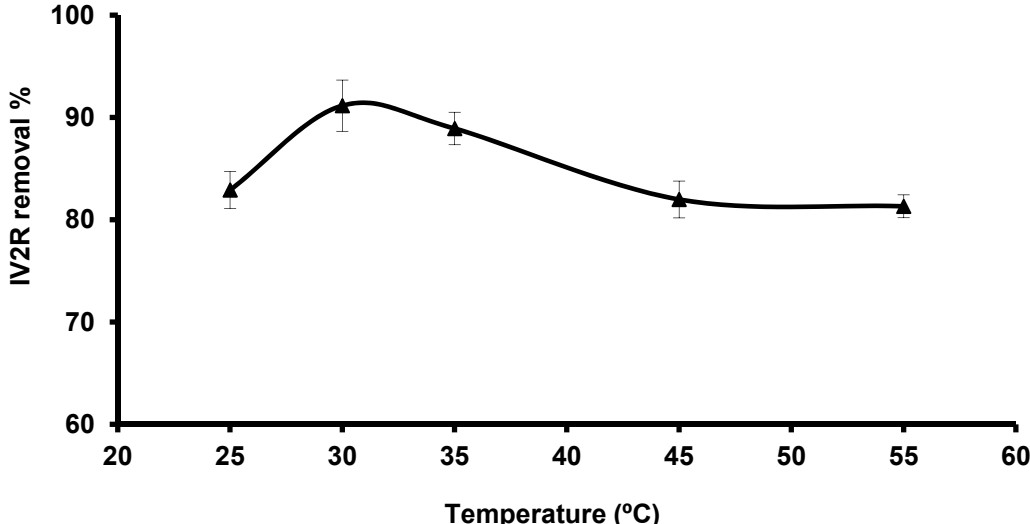

**Figure 10.** Influence of temperature on the removal of IV2R.

### *3.3. Isothermal Analysis*

3.3.1. Freundlich Isotherm

The obtained results were fit with the experimental data of the Freundlich isotherm model, which were confirmed by a high correlation coefficient of $R^2 = 0.989$ for the MWC-NTs, indicating that this model is favorable for the adsorption process (Figure 11). A strong bond occurred between the IV2R dye and the adsorbents, as indicated by the value of $1/n$, which is called the heterogeneity factor, describing the deviation from the linearity of sorption as follows: When $1/n$ is equivalent to 1, the adsorption is linear, and the ammonia particle concentration does not influence the division between the two stages. When $1/n$ is below 1, chemical adsorption occurs; however, when $1/n$ is more than 1, cooperative adsorption occurs, and this adsorption is more favorable physically and includes strong interactions among the particles of the adsorbate [6]. In this work, the value of factor "$1/n$" was less than 1; the data shows chemical sorption method on a surface with this isotherm equation being favorable.

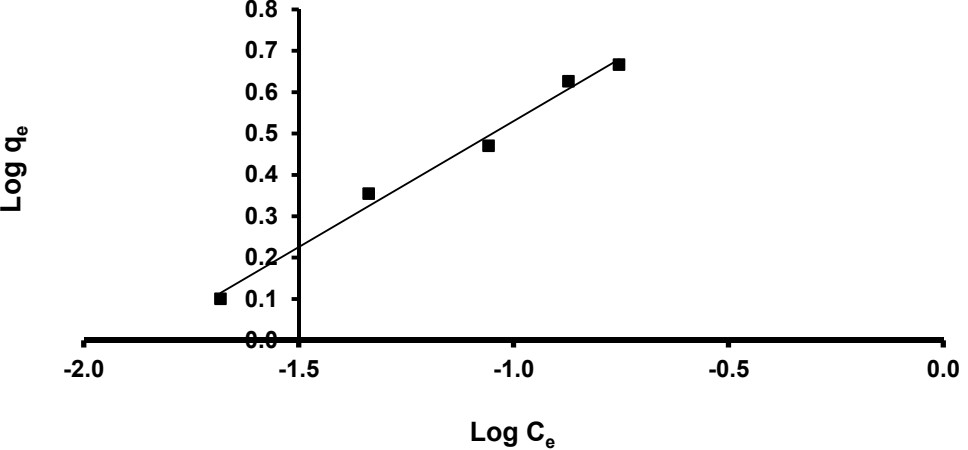

**Figure 11.** Freundlich isotherm of the IV2R onto the MWCNTs.

3.3.2. Langmuir Isotherm

When the Langmuir isotherm was applied, the obtained results agreed with the data throughout the experiment, with a high correlation coefficient of $R^2 = 0.993$ for the MWCNTs. Furthermore, the IV2R dye showed a high maximal uptake capacity ($q_{max}$) of 76.92 mg g$^{-1}$ by the MWCNTs (Figure 12). This is consistent with the formation of a complete monolayer on the surface of the adsorbent. The Langmuir constant (b), which is related to the heat of adsorption, was found to be 1.42.

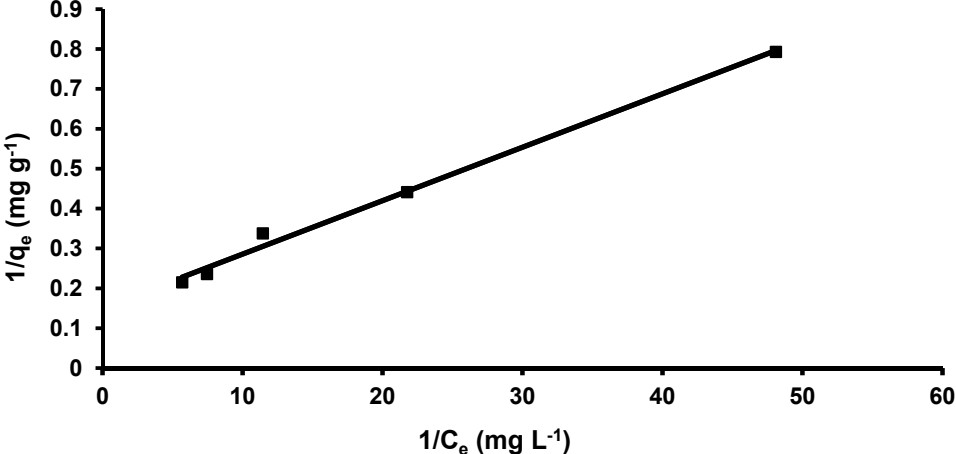

**Figure 12.** Linear Langmuir isotherm for the sorption of IV2R.

### 3.3.3. Halsey and Henderson Isotherm

Halsey and Henderson isotherm equations are suitable for multilayer adsorption; in particular, the fitting of these equations can be heteroporous solids [60]. A plot of $\ln q_e$ versus $\ln C_e$ Halsey and $\ln[-\ln(1-C_e)]$ versus $\ln q_e$. Halsey and Henderson adsorption isotherms are given in Figures 13 and 14, respectively. The isotherm constants and correlation coefficients are summarized in Table 2. Halsey showed a correlation coefficient of $R^2 = 0.989$, while Henderson showed an $R^2$ of 0.988. The results obtained for Halsey and Henderson show that both models are applicable for the adsorption of IV2R onto MWCNTs.

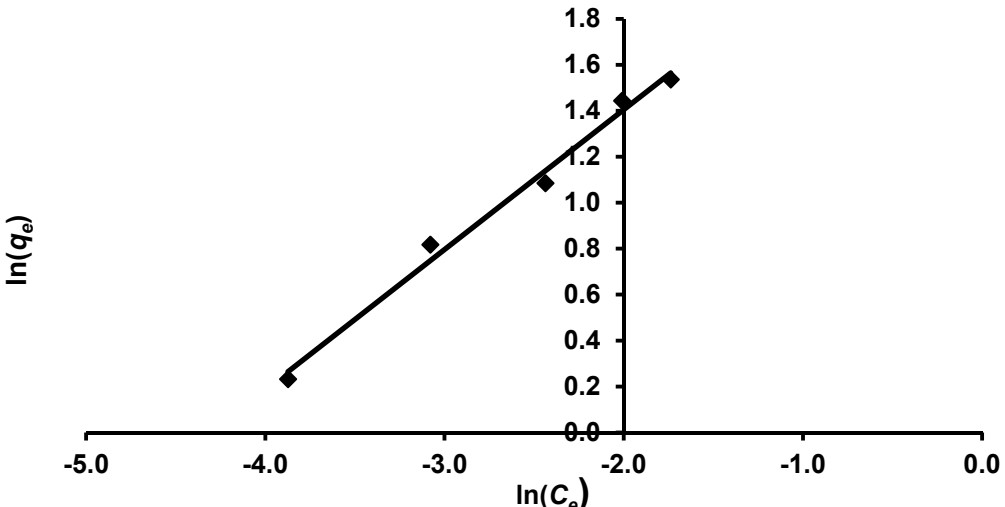

**Figure 13.** Halsey isotherm of the removal of IV2R.

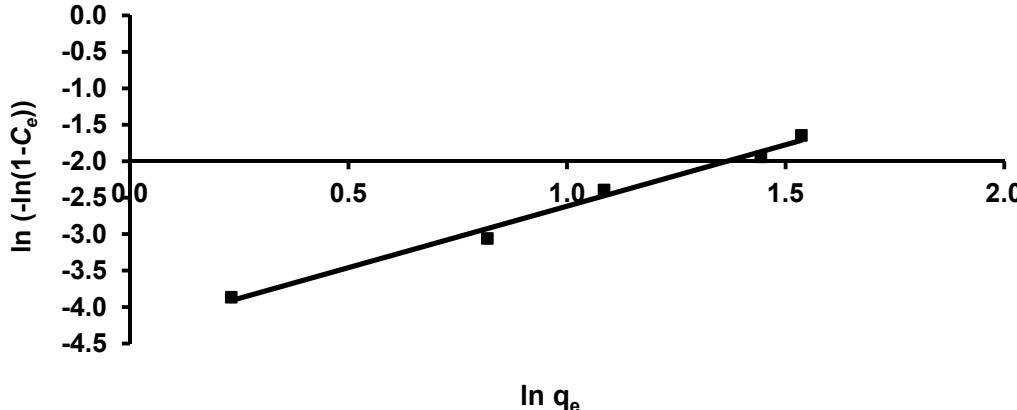

**Figure 14.** Linear Henderson isotherm of the removal of IV2R.

### 3.3.4. Harkins–Jura Isotherm

The Harkins–Jura adsorption isotherm can be expressed as Equation (8), which can be solved by a plot of $1/q_e$ versus $\log C_e$, as shown in Figure 15. The Harkins–Jura model can be described as the presence of a distribution of heterogeneous pores and multilayer adsorption. The isotherm constants and correlation coefficients ranged from $R^2 = 0.858$ (Table 2). This may indicate that the Harkins–Jura model is less applicable.

### 3.3.5. Smith Isotherm

The Smith model is useful for explaining the adsorption isotherm of a biological substance such as cellulose and starch, in addition to being appropriate for heteroporous solids and multilayer adsorption. The Smith model can be solved by a plot of $q_e$ versus $\ln(1-C_e)$, as presented in Figure 16. The Smith model adequately represents the sorption isotherms throughout the entire range of water activity. However, the Smith equation was

successful in describing the isotherms of IV2R onto the MWCNTs, because the model gave a higher $R^2$ value (0.964).

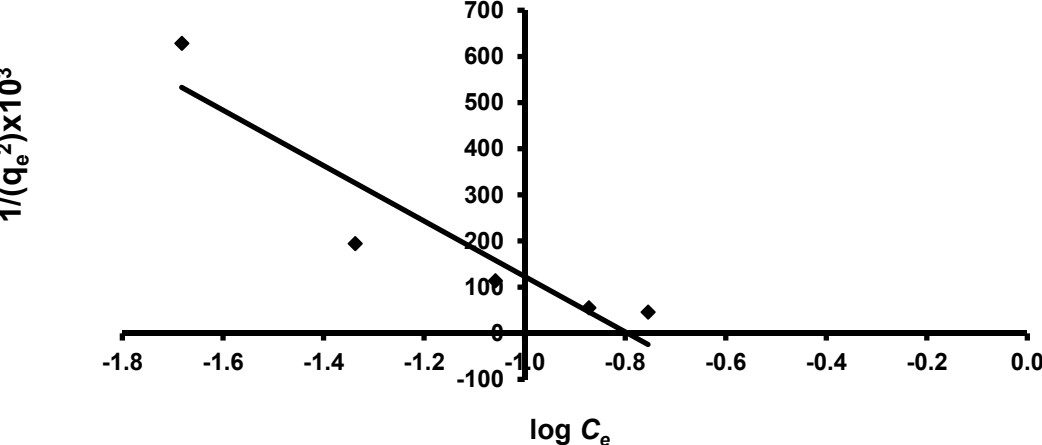

**Figure 15.** Linear Harkins–Jura isotherm of the removal of IV2R.

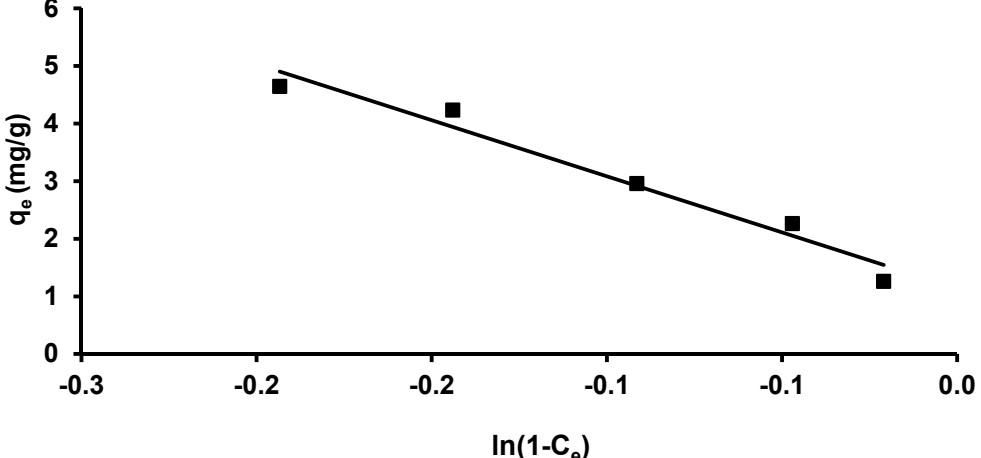

**Figure 16.** Linear Smith isotherm of the removal of IV2R.

### 3.3.6. Tempkin Isotherm

The Tempkin isotherm model was applied. Consequently, linear plots of ($q_e$) vs. (ln $C_e$) allowed for the calculation of the Tempkin isotherm parameters $K_T$ and $b_T$ from the slope and intercept, respectively, as presented in Figure 17 and Table 2. The data indicate that the Tempkin isotherm model applies to the adsorption of IV2R dye onto MWCNTs, as shown by the high linear regression correlation coefficient ($R^2 = 0.965$).

### 3.3.7. Error Function Examination for the Best and Most Appropriate Isotherm Model

The best-fit model for the investigational data was calculated by numerous error functions, such as the average percentage error (APE) equation, chi-square error ($\chi^2$) equation, and the sum of absolute errors (EABS). The data obtained from the different error functions are summarized in Table 3. From the observed data, the best appropriate isotherm models are Tempkin, Smith, Henderson, Freundlich, Langmuir, Halsey, and Harkins–Jura. Nevertheless, the error functions test provided variable data for all models, and the evaluation among the models focused on all error functions individually. For the EABS and $\chi^2$ equation, all the isotherm models can be applied to the investigational data, except the Harkins–Jura isotherm.

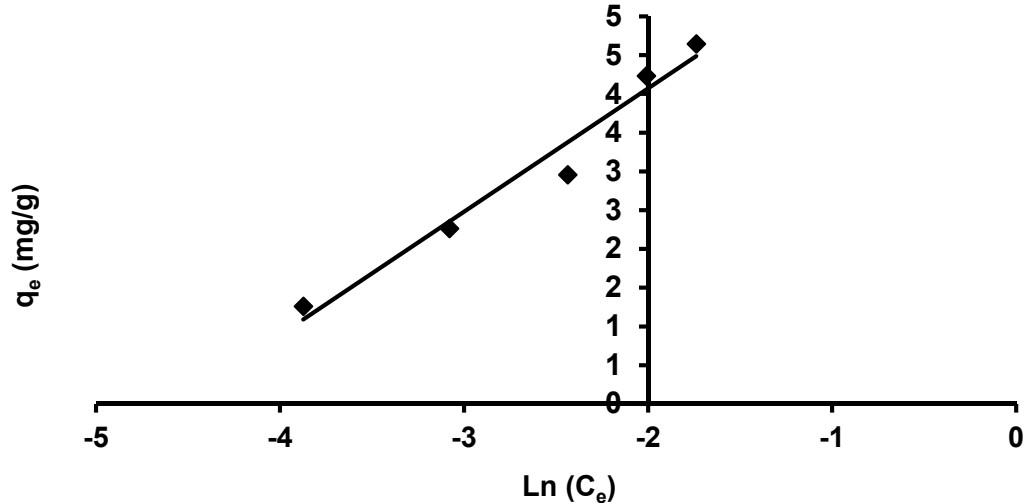

**Figure 17.** Linear Tempkin isotherm of the removal of IV2R.

**Table 2.** Factors of the isotherm models from nonlinear and linear solvation.

| Isotherm Model | Isotherm Parameter | Value |
|---|---|---|
| Freundlich | $1/n$ | 13.77 |
| | $K_F$ $(mg^{1-1/n}L^{1/n}g^{-1})$ | 0.608 |
| | $R^2$ | 0.989 |
| Langmuir | $Q_{max}$ $(mg\ g^{-1})$ | 76.92 |
| | $b$ | 1.42 |
| | $R^2$ | 0.99 |
| Harkins–Jura | $A_{HJ}$ | 1.31 |
| | $B_{HJ}$ | 1.73 |
| | $R^2$ | 0.858 |
| Halsey | $1/n_H$ | 0.6089 |
| | $K_H$ | 2.714 |
| | $R^2$ | 0.989 |
| Henderson | $1/n_h$ | 1.688 |
| | $K_h$ | 0.014 |
| | $R^2$ | 0.988 |
| Smith | $W_{bs}$ | 1.138 |
| | $W_s$ | 19.46 |
| | $R^2$ | 0.964 |
| Tempkin | $A_T$ | 95.58 |
| | $B_T$ | 1.59 |
| | $b_T$ | 1558.2 |
| | $R^2$ | 0.965 |

### 3.4. Adsorption Kinetics

3.4.1. Pseudo-First-Order Kinetics Model

The pseudo-first-order model is the latest well-known equation applied to describe the sorption rate depending on the capacity of adsorption. This model states that the ratio of occupation of adsorption sites is proportional to the number of unoccupied sites [61].

Figure 18A shows the linear figure of log ($q_e$–$q_t$) versus t at the original dye concentration of 10 mg/L. The $q_e$ and $K_1$ values were calculated using the intercept, as well as the slope of the linear plots for removal of the IV2R dye from the MWCNTs.

**Table 3.** The most appropriate isotherm model to investigate the equilibrium data through various errors functions.

| Isotherm Model | APE% | $\chi^2$ | EABS |
|---|---|---|---|
| Freundlich | 0.031 | 0 | 0.034 |
| Langmuir | 2.079 | 0.326 | 2.236 |
| Harkins–Jura | 85.602 | 551.664 | 92.065 |
| Halsey | 12.38 | 11.539 | 13.315 |
| Henderson | 0.022 | 0 | 0.024 |
| Smith | 0.004 | 0 | 0.004 |
| Tempkin | 0.001 | 0 | 0.001 |

### 3.4.2. Pseudo-Second-Order Kinetics Model

The pseudo-second-order equation depends on the assumption of the rate-limiting step due to chemical adsorption containing valence forces by exchange and/or sharing of electrons between dye ions and the carbon nanotube (adsorbent). The pseudo-second-order equation is expressed following [62]. Plots of ($t/q_t$) against ($t$) should offer a linear correlation from which the data of parameters $q_e$ and $k_2$ can be calculated from the slope and intercept, respectively. It can be observed in Figure 18B and Table 4 that this model fits well, with a higher correlation coefficient ($R^2$ = 0.999). Apart from this, it was obvious that the value of the $k_2$ parameter was greater than the corresponding $k_1$ parameter value. This is because the pseudo-second-order model presumes that the adsorption rate is proportional to the square of a number of unoccupied sites, as reported by Joseph and David et al. [53].

Table 4 provides the values of $K_1$, the experimental and calculated values of $q_e$, and the correlation coefficients for the pseudo-first-order kinetics plots. The theoretical values of $q_e$ do not agree with experimental data obtained. This suggests a poor fit between the kinetics data and the pseudo-first-order model. Thus, the pseudo-first-order equation was ruled out as a result of its correlation coefficients ($R^2$) for the current experimental results being small for the IV2R dye in MWCNTs.

**Table 4.** Parameters of the different adsorption kinetics models.

| Kinetic Models | Parameters | Value |
|---|---|---|
| First-order | $q_e$ (calc.) (mg g$^{-1}$) | 29.3 |
| | $k_1 \times 10^3$ (min$^{-1}$) | 6.6787 |
| | $R^2$ | 0.0797 |
| Second-order | $q_e$ (calc.) (mg g$^{-1}$) | 1.066 |
| | $k_2 \times 10^3$ (mg g$^{-1}$ min$^{-1}$) | 9300.677 |
| | $R^2$ | 0.9993 |
| Intraparticle diffusion | $K_{dif}$ (mg g$^{-1}$ min$^{-0.5}$) | 13.434 |
| | $C$ cal (mg g$^{-1}$) | 0.862 |
| | $R^2$ | 0.867 |

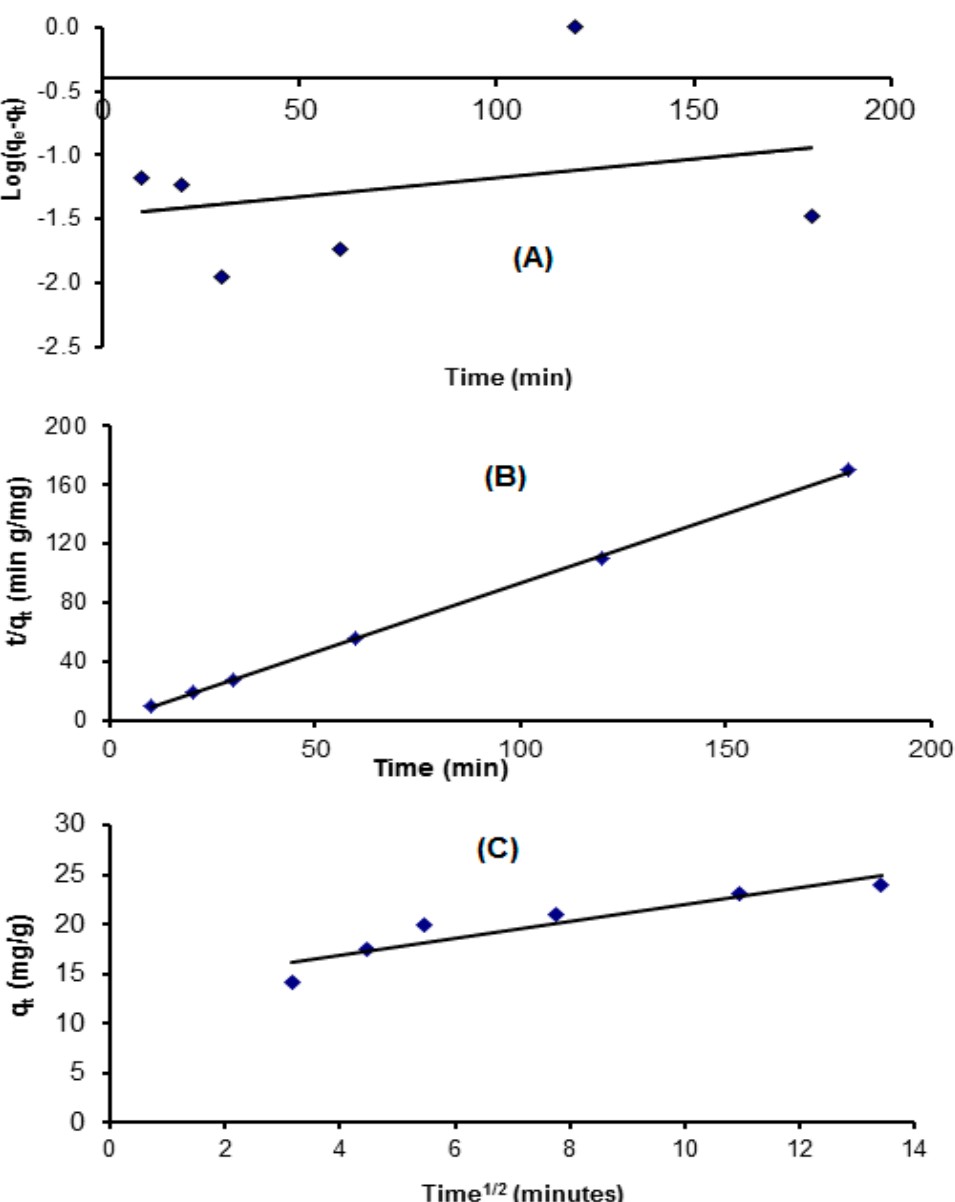

**Figure 18.** Adsorption kinetics of the pseudo-first-order (**A**), pseudo-second-order (**B**), and intraparticle diffusion equation (**C**) of IV2R adsorption.

### 3.4.3. The Intraparticle Diffusion Equation

The adsorption method requires numerous steps, including the transport of solute particles from the aqueous part to the external of the solid molecules, followed by diffusion of the solute molecules inside the interior of the cavities, where is likely to be a slow process and a rate-determining step [15]. The figures of $q_t$ inverse $t^{0.5}$ may present a numerous-linearity correlation, which shows that two or more stages happen through the adsorption process (Figure 18C). The rate constant $K_{dif}$ is directly estimated by the slope and the intercept is $C$, as reported in Table 4. The value of the $C$ factor offers information about the thickness of the border layer, since the resistance to the exterior mass transfer rises as the intercept increases. The low linearity of the plots demonstrates that intraparticle diffusion in the uptake of the IV2R dye onto the MWCNTs with a low correlation coefficient of $R^2 = 0.867$.

### 3.5. Adsorption Thermodynamics

The thermodynamic factors of the IV2R adsorption onto the MWCNTs are given in Table 5. The great negative value of $\Delta G°$ confirms that the sorption of dye was spontaneous and feasible. Nevertheless, the $\Delta G°$ values in Table 5 show that the free energy values increased with decreasing temperature, indicating that the adsorption method is endothermic [37,41] and suggesting that the spontaneity of the adsorption method reduces at a lower temperature. In general, the range of $\Delta G°$ values is between –20 and 0 kJ mol$^{-1}$ for physisorption; however, the chemisorption process is between –80 and –400 kJ mol$^{-1}$ [63]. The observed data of activation energy in this study indicate that the sorption of the IV2R onto the MWCNTs was by physisorption process. The value of $\Delta H°$ for this technique is positive, demonstrating an endothermic nature. The negative sign of the $\Delta S°$ parameter shows an increase in the affinity of the MWCNT for the dye, as well as in randomness at the solid–liquid interface [37].

**Table 5.** Thermodynamic factors of the sorption of IV2R onto MWCNTs.

| Temperature (°C) | $\Delta G°$ (kJ mol$^{-1}$) | $\Delta H°$ (kJ mol$^{-1}$) | $\Delta S°$ (J mol$^{-1}$) |
|---|---|---|---|
| 25 | −7.87669 | | |
| 35 | −7.94698 | 21.877 | −98.76 |
| 45 | −9.763 | | |
| 55 | −10.5637 | | |

### 3.6. Regeneration and Reusability Study

The regeneration and reuse of the sorbent material are essential for its economic viability. To evaluate the reusability of MWCNTs, adsorption–desorption experiments were carried out using 40 mg L$^{-1}$ of dye solution, shaken at 110 rpm for 2 h. The results showed that the removal efficiency of the MWCNTs for IV2R uptake dye remained nearly unchanged for the three consecutive series. The initial removal efficiency was 95.2% in 120 min, which reduced slightly to 92.1% after the third round, as shown in Figure 19. Hence, it could be concluded that the sorption capacity of MWCNTs remains unaffected with extended regeneration rounds. This decrease shows that the adsorbent represents a good material that can be used in industrial operations. The diminishing adsorption of IV2R dyes over the three cycles was brought about by the destructive influence of stripping agents and the weight loss in the adsorbent [64]. Furthermore, the resident IV2R dyes in the adsorbent (irreversible binding) resulted in a decrease in the number of available binding sites [65,66].

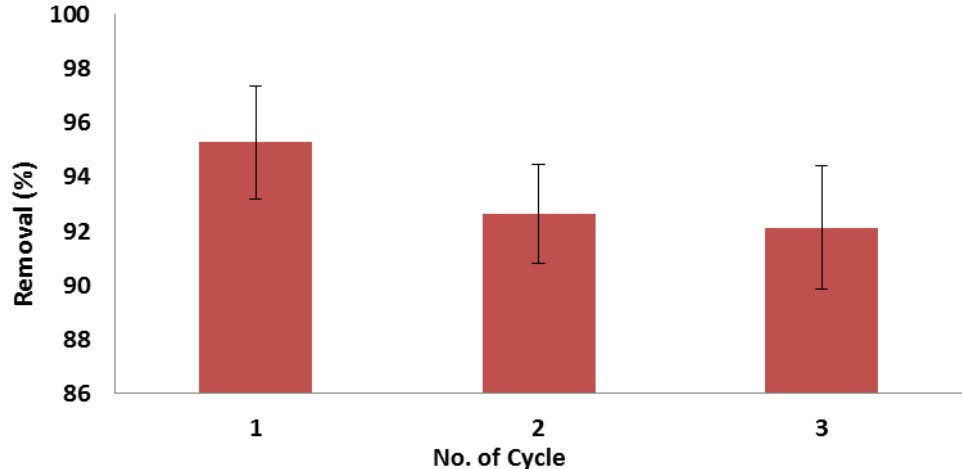

**Figure 19.** Reusability study for removal of IV2R in various cycles.

Ahmad et al. [67] achieved more than 85% removal of dyes, indicating the reusability of the adsorbent. Moreover, Tunali et al. [68] tested the regenerated adsorbent for up to five adsorption–desorption cycles and showed that the decline in efficiency was less than 5%, highlighting the favorable prospects of these fungi for Pb(II) elimination in industrial operations. Gupta et al. [69] studied the regeneration of MWCNTs for reactive red 3BS, and 92.8% reuse efficiency was achieved after four cycles.

### 3.7. Applicability on Actual Wastewater

To evaluate the validity of the use of MWCNTs as adsorbents, real wastewater mixed with simulated dye samples was collected to examine the removal of IV2R by the adsorbent under optimization conditions. The results confirmed that the highest removal was significantly altered by changing the kind of water, for which deionized water exhibited the lowest influence on the adsorption method, with color removal of 98.4% at an acidic pH after 120 min. By contrast, real wastewater comprises very high concentrations of interfering ions from numerous pollutants, which had a significant influence on the elimination efficiency of IV2R dye, with color removal of 88% after 120 min. Regardless, the results revealed that MWCNTs adsorbents can be successfully used as a low-cost adsorbent to remove IV2R dye from aqueous mixtures and wastewater. The process showed a typical feature of effluents after the treatment of dye, observed to be 110 mg L$^{-1}$ of TSS and 652 mgL$^{-1}$ of TDS, as presented in Table 6. Moreover, these results show that wastewater containing dye is well treatable through this method, with percentages of color removal of 98.4% and 88.2%; additionally, TSS and TDS removal decreased after the treatment process.

**Table 6.** Characteristics of the actual wastewater before and after treatment, and standards for the safe disposal of prescribed dye effluents.

| Parameters | Before Treatment | After Treatment | Standards for Cotton Textile Industries [66] |
|---|---|---|---|
| pH | 9.5 | 2–6 | 5.5–9.0 |
| TSS (mg L$^{-1}$) | 2050 | 110 | 100 |
| TDS (mg L$^{-1}$) | 3240 | 652 | 500 |

### 3.8. Comparative Studies of the Sorption Capacity of MWCNTs

The results of the present study were compared to those of other works on the adsorption capacities of various dyes to show the efficiency of MWCNTs (Table 7).

### 3.9. Future Research Perspectives and Hallenges

Although a lot of studies have successfully examined the treatment efficiency of MWCNTs and CNT-based adsorbents, more studies should be focused on the following:

(a) Improving CNT filters, films, and sheets for real industrial wastewater purification on an experimental scale.
(b) Creating purified and functionalized compounds of CNT in commercial amounts with little environmental impact at a reasonable price.
(c) Producing CNTs with comparable adsorption capabilities through various techniques such as laser ablation, chemical vapor deposition, and arc discharge processes.
(d) Predicting the adsorption mechanism and dye elimination capability from real industrial wastewater under a range of operating conditions in batch and column processes, as well as on a larger scale.
(e) Examining the toxicity of CNTs to the environment.

**Table 7.** Summary of the elimination of several dyes from aquatic mixtures by various carbon nanotubes.

| CNTs | Dye Adsorbed | $q_e$ (mg g$^{-1}$) | Ref. |
|---|---|---|---|
| MWCNTs | Sufranine O | 43.48 | [65] |
| MWCNTs | Methylene blue | 35.4 | [67] |
| Oxidize MWCNTs | Bromothymol blue | 55 | [25] |
| SWCNT–COOH | Basic red 46 | 45.33 | [1] |
| MWCNTs | methylene blue | 64.7 | [67] |
| MWNTs | Orange II | 66.12 | [68] |
| SWCNT | Basic red 46 | 38.35 | [69] |
| MWNTs | Reactive blue | 335.7 | [19] |
| MWNTs | Alizarin red S | 161.290 | [70] |
| MWCNTs | Reactive blue 4 | 502.5 | [19] |
| MWNTs | Methyl orange | 52.86 | [17] |
| CNTs/activated carbon fiber | Basic violet 10 | 220 | [71] |
| MWCNTs/Fe$_3$C | Direct Red 23 | 172 | [72] |
| Chitosan/Fe$_2$O$_3$/MWCNTs | Methyl orange | 66.90 | [21] |
| MWCNTs/Fe$_2$O$_3$ | Methylene blue | 42.3 | [73] |
| MWCNT | | 76.9 | Present study |

## 4. Conclusions

MWCNTs were prepared, purified, and functionalized in this study. MWCNTs are good alternative adsorbents with which to remove IV2R dye from aqueous solutions and wastewater. MWCNT nanocompounds were examined using SEM, FTIR, XRD, and BET surface analysis. Purified MWCNTs were characterized by FTIR, which showed the existence of functional groups such as N-H, O-H, COOH, C=N, and S-H stretching, thereby increasing their ion exchange properties for the selective adsorption of opposing charged molecules. Furthermore, various factors affecting IV2R dye elimination efficiency were examined by the batch process, including adsorbent dose, agitation time, pH, initial dye concentration, and temperature. The highest operation influences were found to be 0.04 g of MWCNTs, 10 mg L$^{-1}$ of dye, pH 4 at 30 °C, and 120 min of contact time. The percentage removal of the IV2R dyes increased with a rise in the contact time and MWCNT dosage, while it decreased with augmented initial dye concentrations, pH, and temperature. On the contrary, the equilibrium isotherm adsorption results were examined using numerous applied isotherms such as the Freundlich, Langmuir, Harkins–Jura, Halsey, Henderson, Smith, and Tempkin models. A higher adsorption capacity ($q_{max}$) of 76.92 mg g$^{-1}$ was acquired from the Langmuir model. The best-fitting isotherm model in this study was obtained by different error function models. However, a kinetics study was conducted using the intraparticle diffusion, pseudo-first-order, and pseudo-second-order models. The regression data indicated that the sorption kinetics equation was the best via the pseudo-second-order model ($R^2$ = 0.999). Additionally, the percentage of removal of the MWCNTs from real wastewater reached 88%. Moreover, the thermodynamic parameters of the sorption process ($\Delta G°$, $\Delta H°$, and $\Delta S°$) were calculated. The adsorption of IV2R dye was endothermic, and was a spontaneous process, and the reaction of adsorption is a physisorption process. The efficiency of the regenerated MWCNTs showed a slight change after three cycles of reuse and regeneration.

**Author Contributions:** Conceptualization, A.E.A.; methodology, A.E.A. and K.M.A.; software, A.E.A. and K.M.A.; formal analysis, A.E.A., M.A., and A.T.M.; investigation, A.E.A.; resources, A.E.A. and K.M.A.; data curation, A.E.A., M.A. and A.T.M.; writing—original draft preparation, A.E.A., M.A., and A.T.M.; writing—review and editing, A.E.A., M.A., K.M.A., and A.T.M.; visualization, K.M.A.

and M.A.; supervision, K.M.A. project administration, A.E.A.; funding acquisition, K.M.A. All authors have read and agreed to the published version of the manuscript.

**Funding:** The authors appreciated Taif University Researchers Supporting Project number TURSP-2020/267, Taif University, Taif, Saudi Arabia.

**Institutional Review Board Statement:** Not applicable.

**Informed Consent Statement:** Not applicable.

**Data Availability Statement:** All relevant data are within the paper, and those are available from the corresponding author.

**Acknowledgments:** The authors appreciated Taif University Researchers Supporting Project number TURSP-2020/267, Taif University, Taif, Saudi Arabia.

**Conflicts of Interest:** The authors declare no conflict of interest.

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
