# Peer review of "Influencing Multi-Walled Carbon Nanotubes for the Removal of Ismate Violet 2R Dye from Wastewater: Isotherm, Kinetics, and Thermodynamic Studies"

_applsci, doi:10.3390/app11114786_

Round 1
Reviewer 1 Report
I read an interesting research article entitled Influencing Multi-Walled Carbon Nanotubes for Efficient Removal of Ismate violet 2R Dye from Wastewater; Isotherm, Kinetic and Thermodynamic studies to Applied Sciences.
The concept of the manuscript is novel, fits and suitable to publish in to Applied Sciences. This manuscript is generally well written and clearly presented however still needs to address many comments, and thus require substantial major revision before its acceptance.
- Title need to be modified which can describe the whole research work. Replace efficient word
- In abstract Remove the introductory part. authors should mention the values of results and importance of research work in one or two sentences. Don’t use any abbreviation here
- Provide a nice graphical abstract representing the research work.
- In the introduction section, write the novelty of the work and the problem statement clearly. For the textile dyes quantitative data and their toxic effects refer and cite important review article Journal of Environmental Chemical Engineering, 105012, 2021. The detailed discussion about the novelty, significance of your research work and research gap relative to the literature is essential.
- Authors forgot to mention biological degradation and an immobilized system for the treatment of dye containing wastewater degradation refer and cite International Biodeterioration & Biodegradation 65 (3), 494-503, 2011. Give detailed research objectives at the end of introduction. Try to avoid cluster of references
- After three repetitive cycle why efficiency is very less give substantial discussion for the same.
- How authors determined the characteristics of used actual wastewater. After treatment what are the changes in other parameters give details?
- This manuscript lacked substantial discussion of results with the literature authors should concentrate on this during revision.
- Write the practical applications and future research perspectives and challenges by adding a new section before conclusions
- The conclusion of the study is not discussed with the specific output obtained from the study, it could be modified with precise outcomes with a take home message.
- English and grammar mistakes are present. The author should check the manuscript by native English Speaker to improve the quality of the manuscript.
Author Response
SUMMARY OF AUTHOR(S) RESPONSE TO REVIEWER’S COMMENTS
Manuscript ID: applsci 128376
Manuscript Title: Influencing Multi-Walled Carbon Nanotubes for Efficient Removal of Ismate violet 2R Dye from Wastewater; Isotherm, Kinetic and Thermodynamic studies.
Authors: Khamael M. Abualnaja, Ahmed E. Alprol,*, Mohamed Ashour, Abdallah Tageldein Mansour
Reviewer 1# round 1 Comment |
Author(s) response |
Major Comments: |
|
I read an interesting research article entitled “Influencing Multi-Walled Carbon Nanotubes for Efficient Removal of Ismate violet 2R Dye from Wastewater; Isotherm, Kinetic and Thermodynamic studies” to Applied Sciences. The concept of the manuscript is novel, fits and suitable to publish in to Applied Sciences. This manuscript is generally well written and clearly presented however still needs to address many comments, and thus require substantial major revision before its acceptance. |
- The authors would like to thank Reviewer # 1 for his kind and his interesting and valuable comments. All Reviewer # 1 comments were considered carefully by the authors. These comments significantly improved the manuscript. |
Title need to be modified which can describe the whole research work. Replace efficient word |
- Title has been modified as the reviewer suggests, and the word “Efficient” has been deleted. (Page: 1, Line: 1). |
In abstract remove the introductory part. Authors should mention the values of results and importance of research work in one or two sentences. Don’t use any abbreviation here. |
- The introductory part has been removed and, in general, the abstract has been improved (Page: 1, Lines: 15 - 32). |
Provide a nice graphical abstract representing the research work. |
- Thank you for this valuable comment. A nice graphical abstract has been provided. |
In the introduction section, write the novelty of the work and the problem statement clearly. For the textile dyes quantitative data and their toxic effects refer and cite important review article Journal of Environmental Chemical Engineering, 105012, 2021. The detailed discussion about the novelty, significance of your research work and research gap relative to the literature is essential. |
- The introduction has been supported by recent articles, in addition, the research problem statement has been added. - (Page: 1, Lines: 40-43), (Page: 2, Lines: 44 - 45; 54 - 56; 64 - 66; and 83 – 92). - The mentioned interesting review article of Kishor et al. 2021 “Journal of Environmental Chemical Engineering, 105012, 2021, https://doi.org/10.1016/j.jece.2020.105012) has been cited in the introduction section (Ref. 5). - The novelty of the work has been added (Page: 3, Lines: 107-112). |
Authors forgot to mention biological degradation and an immobilized system for the treatment of dye containing wastewater degradation refer and cite International Biodeterioration & Biodegradation 65 (3), 494-503, 2011. |
- This interesting article (Saratale et al. 2011, https://doi.org/10.1016/j.ibiod.2011.01.012) has been provided (Ref. 4) in the introduction section (Pages: 1 and 2, Lines: 43 – 45; and 50 – 54).
|
Give detailed research objectives at the end of introduction. Try to avoid cluster of references |
- The research objectives have been detailed at (Page: 3, Line 96-112). |
After three repetitive cycles why efficiency is very less give substantial discussion for the same. |
- The discussion has been improved (Page: 22 & 23, Lines: 675-680; and 683 – 688). |
How authors determined the characteristics of used actual wastewater. After treatment what are the changes in other parameters give details? |
- Determination the characteristics of used actual wastewater has been showed and added (Page: 8, Lines: 355 – 358 & 362 - 365). - The change in other parameters after treatment has been added with additional results (Page: 23, Lines: 696-697; 694; 698-705). |
This manuscript lacked substantial discussion of results with the literature authors should concentrate on this during revision. |
- Further discussion with the literature authors was added in manuscript. (Page: 12, Lines: 449– 452 & 465-468) & (Page: 13, Lines: 478 & 485-488) & (Page: 14, Lines: 500-503). |
Write the practical applications and future research perspectives and challenges by adding a new section before conclusions |
- *Added in manuscript at section 3.8. - 3.8. Future research perspectives and challenges (Page: 24, Lines: 712-725). |
The conclusion of the study is not discussed with the specific output obtained from the study; it could be modified with precise outcomes with a take home message. |
- The conclusion section has been improved, revised, rewritten, and supported with the principal results, and supported by the obtained results (Page: 24, Lines: 727 – 751). |
English and grammar mistakes are present. The author should check the manuscript by native English Speaker to improve the quality of the manuscript. |
- The manuscript was check by native English Speaker from authorized center of MDPI. |
We would like to extend our sincere thanks and appreciation to the reviewers and editorial board. In fact, their comments and guidance added a lot to the research and increased its scientific content. Therefore, the words cannot express their gratitude for their time and effort they put in evaluating this research.

Reviewer 2 Report
This study “applsci-1218376, Influencing Multi-Walled Carbon Nanotubes for Efficient Removal of Ismate violet 2R Dye from Wastewater; Isotherm, Kinetic and Thermodynamic studies” investigates multiwall carbon nanotubes (MWCNT), as a novel approach, and application of MWCNT as adsorption substance to remove Ismate violet 2R (IV2R) dye from aquatic effluents under various conditions.
This is a very nice publication carefully put together, well written, and well explained, with lots of work and experiments having been done, and lots of useful data. A very complete manuscript. However, the structure is confusing, this manuscript has several parts and tests bud there is no cohesion. Authors should connect all of the sections together.
- The language used in the introduction can be more specific to the scope and aim of the study.
- The conclusion section is very long and general. Must be rewritten and revised.
Author Response
SUMMARY OF AUTHOR(S) RESPONSE TO REVIEWER’S COMMENTS
Manuscript ID: applsci 128376
Manuscript Title: Influencing Multi-Walled Carbon Nanotubes for Efficient Removal of Ismate violet 2R Dye from Wastewater; Isotherm, Kinetic and Thermodynamic studies.
Authors: Khamael M. Abualnaja, Ahmed E. Alprol,*, Mohamed Ashour, Abdallah Tageldein Mansour
Reviewer 2# round 1 Comment |
Author(s) response |
Comments: |
|
This study “applsci-1218376, Influencing Multi-Walled Carbon Nanotubes for Efficient Removal of Ismate violet 2R Dye from Wastewater; Isotherm, Kinetic and Thermodynamic studies” investigates multiwall carbon nanotubes (MWCNT), as a novel approach, and application of MWCNT as adsorption substance to remove Ismate violet 2R (IV2R) dye from aquatic effluents under various conditions. This is a very nice publication carefully put together, well written, and well explained, with lots of work and experiments having been done, and lots of useful data. A very complete manuscript. However, the structure is confusing, this manuscript has several parts and tests bud there is no cohesion. Authors should connect all of the sections together. |
|
The language used in the introduction can be more specific to the scope and aim of the study. |
- The introduction has been supported by recent articles, in addition, the research problem statement has been added. - (Page: 1, Lines: 40-43), (Page: 2, Lines: 44 - 45; 54 - 56; 64 - 66; and 83 – 92). - The novelty of the work has been added (Page: 3, Lines: 107-112). |
The conclusion section is very long and general. Must be rewritten and revised. |
The conclusion section has been improved, revised, rewritten, and supported with the principal results, and supported by the obtained results (Page: 24, Lines: 727 – 751). |
English and grammar mistakes are present. The author should check the manuscript by native English Speaker to improve the quality of the manuscript. |
-The manuscript was check by native English Speaker from authorized center of MDPI. |
We would like to extend our sincere thanks and appreciation to the reviewers and editorial board. In fact, their comments and guidance added a lot to the research and increased its scientific content. Therefore, the words cannot express their gratitude for their time and effort they put in evaluating this research.

Reviewer 3 Report
This paper considers the application of multiwall carbon nanotubes as adsorption material for Ismate violet dye removal from aquatic effluents under various conditions. The submitted article is interesting, original and within the scope of the journal. In addition, the manuscript is well illustrated and presents in a scientific manner the subject, but some minor changes should be addressed:
- The authors mentioned about various treatment wastewater methods (from line 47 to 50), please come with several references at the end of the comment (please see https://doi.org/10.3390/nano10030474, DOI: 10.5593/sgem2017/31/S12.058 and DOI: 10.5593/sgem2017H/33/S12.020). It will be also interesting to discuss briefly about these methods (advantages and disadvantages) in order to highlight your choose for dye removal from aquatic effluents.
- Please remove the gridlines from figures.
- Please rewrite the axes title and scales in figure 2 in order to be more readable.
- Please make de SEM parameters (in the bottom of micrographs) more readable.
- I recommend to use the same scale for SEM examination.
- In my opinion are too many figures, try to merge them.
Author Response
SUMMARY OF AUTHOR(S) RESPONSE TO REVIEWER’S COMMENTS
Manuscript ID: applsci 128376
Manuscript Title: Influencing Multi-Walled Carbon Nanotubes for Efficient Removal of Ismate violet 2R Dye from Wastewater; Isotherm, Kinetic and Thermodynamic studies.
Authors: Khamael M. Abualnaja, Ahmed E. Alprol,*, Mohamed Ashour, Abdallah Tageldein Mansour
Reviewer 3# round 1 Comment |
Author(s) response |
Comments: |
|
This paper considers the application of multiwall carbon nanotubes as adsorption material for Ismate violet dye removal from aquatic effluents under various conditions. The submitted article is interesting, original and within the scope of the journal. In addition, the manuscript is well illustrated and presents in a scientific manner the subject, but some minor changes should be addressed: |
- The authors would like to thank Reviewer # 3 for his kind and his interesting and valuable comments which significantly improved the manuscript. -The manuscript was check by native English Speaker from authorized center of MDPI. |
The authors mentioned about various treatments wastewater methods (from line 47 to 50), please come with several references at the end of the comment (please see https://doi.org/10.3390/nano10030474, DOI: 10.5593/sgem2017/31/S12.058, and DOI: 10.5593/sgem2017H/33/S12.020). It will be also interesting to discuss briefly about these methods (advantages and disadvantages) in order to highlight your choose for dye removal from aquatic effluents. |
- These three interesting references have been added and cited in the manuscript (Ref. 8 – 10). - Disadvantages of these techniques have been added (Page: 2, Lines: 55 – 56). - The advantages of adsorption technology were showed in (Page: 2, Lines: 60 – 66). - While, the advantages of MWCNTs have been presented (Page: 2, Lines: 73 – 75). - The introduction has been supported by recent articles; in addition, the research problem statement has been added. (Page: 1, Lines: 40-43), (Page: 2, Lines: 44 - 45; 54 - 56; 64 - 66; and 83 – 92). -The novelty of the work has been added (Page: 3, Lines: 107-112). |
Please remove the gridlines from figures. |
- The gridlines was removed from all figures. |
Please rewrite the axes title and scales in figure 2 in order to be more readable. |
- The axes title and scales of Figure 2 has been rewrite. |
Please make de SEM parameters (in the bottom of micrographs) more readable. |
- SEM parameters were added in the caption of Figure 3.
|
I recommend to use the same scale for SEM examination. |
- Figure 3A has been added at 2 µm to use the same scale for SEM examination |
In my opinion are too many figures, try to merge them. |
- Some Figures have been merged in Figure 18. |
We would like to extend our sincere thanks and appreciation to the reviewers and editorial board. In fact, their comments and guidance added a lot to the research and increased its scientific content. Therefore, the words cannot express their gratitude for their time and effort they put in evaluating this research.

Round 2
Reviewer 1 Report
Authors have substantially revised the manuscript according to the comments.
The present form of the manuscript can be accepted for publication.
Reviewer 2 Report
The paper has been improved and corresponding modifications have been conducted. I think the current version can be considered for publication.